# Reference Gene Selection for RT-qPCR Normalization in *Toxoplasma gondii* Exposed to Broxaldine

**DOI:** 10.3390/ijms252111403

**Published:** 2024-10-23

**Authors:** Yanhua Qiu, Yubin Bai, Weiwei Wang, Qing Wang, Shulin Chen, Jiyu Zhang

**Affiliations:** 1Key Laboratory of New Animal Drug Project of Gansu Province, Lanzhou 730050, China; qyhranglin@163.com (Y.Q.); baiyb1011@163.com (Y.B.); weiweiwang1990@163.com (W.W.); qingwang880826@163.com (Q.W.); 2Key Laboratory of Veterinary Pharmaceutical Development, Ministry of Agriculture and Rural Affairs, Lanzhou 730050, China; 3Lanzhou Institute of Husbandry and Pharmaceutical Sciences, Chinese Academy of Agricultural Sciences, Lanzhou 730050, China; 4College of Veterinary Medicine, Northwest Agriculture & Forestry University, Yangling 712100, China

**Keywords:** *Toxoplasma gondii*, broxaldine, gene stability, reference genes, RT-qPCR

## Abstract

Reverse transcription–quantitative real-time polymerase chain reaction (RT-qPCR) is widely used to accurately assess target gene expression. Evaluating gene expression requires the selection of appropriate reference genes. To identify reliable reference genes for *Toxoplasma gondii* (*T. gondii*) under varying concentrations of broxaldine (BRO), we employed the ΔCt method, BestKeeper, NormFinder, GeNorm, and the comprehensive web-based platform RefFinder to assess the expression stability of ten candidate reference genes in *T. gondii*. Herein, our findings reveal that the stability of these candidate reference genes is influenced by different experimental conditions. Under normal conditions, the most stable genes were *TGME49_205470* and *TGME49_226020*. However, the most stable genes differed when BRO concentrations were at 1, 2, and 4 μg/mL. Across all samples, *TGME49_247220* and *TGME49_235930* were identified as the most stable reference genes. Moreover, we also confirmed the stability of *TGME49_247220* and *TGME49_235930* as reference genes through RT-qPCR assays. The present study provides a foundation for applying the RT-qPCR method to investigate target gene expression following BRO treatment in *T. gondii*.

## 1. Introduction

Reverse transcription–quantitative real-time polymerase chain reaction (RT-qPCR) has become an effective tool for assessing mRNA levels due to its high reproducibility, strong specificity, and capacity for high throughput [1]. Nevertheless, several key principles must be followed to ensure reliable results. These include using high-quality RNA and using primers with strong specificity and good amplification efficiency, as well as selecting stable mRNA reference genes to ensure accurate data correction and standardization [2,3]. In RT-qPCR experiments, researchers typically select widely used reference genes for normalizing gene expression [4]. However, numerous studies have shown that these commonly used reference genes, such as *GAPDH* and *β-Actin*, display variable expression levels under different conditions [5,6,7].

*Toxoplasma gondii* (*T. gondii*) is one of the most successful parasites worldwide [8]. Of note, acute infections can be fatal for individuals with compromised or weakened immune systems, while chronic infections have been linked to various neurological disorders [9]. Toxoplasmosis remains a significant global public health challenge, with effective chemotherapy being the primary strategy for combating this disease [10]. Our previous studies have shown that broxaldine (BRO) has beneficial effects on both *T. gondii* tachyzoites and bradyzoites, potentially influencing the organism’s autophagy, mitochondrial dysfunction, and neutral lipid synthesis [11]. Nonetheless, the precise mechanism by which BRO affects *T. gondii* remains elusive. RNA-seq data following treating *T. gondii* with BRO suggest that BRO may influence the expression of commonly used reference genes in *T. gondii*. Therefore, to improve the reliability of gene expression analysis, it is crucial to investigate the selection of reference genes in *T. gondii* in response to varying concentrations of BRO.

The present study evaluated the stability of ten candidate reference genes in *T. gondii* under varying concentrations of BRO. To assess the reliability and accuracy of these reference genes, the expression patterns of fifteen genes were analyzed under different experimental conditions involving BRO, utilizing transcriptome data and findings from previous research.

## 2. Results

### 2.1. Primer Specificity and Amplification Efficiency

Total RNA was extracted from *T. gondii* treated with BRO at concentrations of 4, 2, and 1 μg/mL. Analysis using 1% agarose gel electrophoresis showed a distinct band in the total RNA sample from *T. gondii*, with no evidence of degradation or contamination (Appendix A). The measured OD260/OD280 ratios were approximately 2.0, indicating that the RNA’s concentration, purity, and integrity met the established standards.

The RT-qPCR results showed that, under varying concentrations of BRO, all candidate reference genes produced a single band of the expected size on a 1% agarose gel, and the melting curves displayed a single peak (Appendix A), indicating good primer specificity. In addition, the correlation coefficients of the standard curves for the candidate reference genes were high (R^2^ > 0.990). Amplification efficiency was calculated using the formula E = (10^(−1/slope) − 1) × 100%. All ten candidate reference genes demonstrated satisfactory amplification efficiency, as illustrated in Appendix A and Table 1.

### 2.2. Candidate Reference Gene Expression Analysis

The Ct value serves as an inverse indicator of gene expression, with lower Ct values indicating higher gene expression levels [12]. We analyzed the expression levels of ten candidate reference genes using RT-qPCR (Figure 1). The results showed that the Ct values of the candidate reference genes ranged from 19.6 to 31.5. Notably, *TGME49_289690* (*GAPDH1*) had the lowest Ct value across varying concentrations of BRO, suggesting that *GAPDH1* exhibited the highest expression level.

### 2.3. Candidate Reference Gene Expression Stability Analysis

#### 2.3.1. ∆Ct Analysis

The stability of candidate reference genes was evaluated using ΔCt analysis, with the gene exhibiting the lowest mean standard deviation (mSD) considered the most stable [13]. The analysis identified *TGME49_205470* as the most stable gene within the control group, while *TGME49_247220* emerged as the most stable gene across all samples. Conversely, *TGME49_316400* was found to be the least stable gene (Table 2 and Figure 2).

#### 2.3.2. BestKeeper Analysis

In the BestKeeper analysis, a smaller standard coefficient of variation (SD) of the Ct value indicates greater stability in gene expression, with genes showing an SD greater than 1 considered unstable [14]. Our findings demonstrate that ten candidate genes exhibited stable expression (SD < 1) in both the control group and the BRO 1 μg/mL and BRO 4 μg/mL groups. However, in the BRO 2 μg/mL group, only *TGME49_247220*, *TGME49_205470*, and *TGME49_235930* displayed stable expressions (Table 3). Across all samples, *TGME49_212300* emerged as the most stable gene, while *TGME49_316400* was the least stable (Table 3 and Figure 2).

#### 2.3.3. NormFinder Analysis

In the NormFinder analysis, a lower stability value indicates greater gene stability. Our results show that *TGME49_205470* is the most stable gene in the control group. Among all samples analyzed, *TGME49_209030* emerged as the most stable gene overall, while *TGME49_316400* was identified as the least stable gene (Table 4 and Figure 2).

#### 2.3.4. GeNorm Analysis

Next, we used geNorm to evaluate gene expression stability across different samples. Candidate genes were ranked based on their average expression M value, with genes exhibiting M < 1.5 considered suitable for normalization analysis [15]. A lower M value indicates greater stability in gene expression. In the control group, the M values of candidate genes ranged from 0.01 to 0.04, while in the BRO group, the values ranged from 0.06 to 0.26. This suggests that gene expression was more stable in the control group compared to the BRO group (Figure 3A–D). Among all samples, *TGME49_247220* and *TGME49_235930* showed the highest stability in expression, whereas *TGME49_316400* demonstrated the least stability (Figure 3E).

In addition to assessing the expression stability of candidate reference genes, geNorm can identify the optimal number of reference genes by analyzing pairwise variation (V_n_/V_n+1_) [15,16]. The results of this study show that across various concentrations of BRO, the V_2/3_ ratio for each gene in all samples consistently remained below the threshold of 0.15 (Figure 4). This finding indicates that two reference genes are sufficient to achieve optimal performance in gene expression analysis.

#### 2.3.5. RefFinder Analysis

Finally, we used RefFinder to conduct a comprehensive analysis of the four methods. The results showed that, in the control group, *TGME49_205470* exhibited the highest stability. In the BRO 1 μg/mL group, *TGME49_247220* was the most stable. In the BRO 2 μg/mL group, *TGME49_226020* demonstrated the greatest stability. In the BRO 4 μg/mL group, *TGME49_205470* again showed the highest stability. Across all samples, *TGME49_247220* and *TGME49_235930* were identified as the most stable genes, while *TGME49_316400* was found to be the least stable (Figure 5).

### 2.4. Verification of Reference Genes

We combined transcriptome data and selected six genes to validate the stability of the reference genes *TGME49_247220* and *TGME49_235930*. The results indicated that the relative expression levels of *TGME49_245980*, *TGME49_273130*, *TGME49_268970*, *TGME49_247520*, *TGME49_260190*, and *TGME49_252360* were consistent with the transcriptome data (Figure 6). Additionally, BRO influenced the expression of autophagy, mitochondrial, and lipid-related genes in *T. gondii* (Figure 7), which is consistent with our previous studies [11]. In summary, our results suggest that *TGME49_247220* and *TGME49_235930* can be used as stable reference genes for RT-qPCR experiments involving BRO.

## 3. Discussion

Studies have shown that reference gene expression levels can fluctuate under various experimental conditions [13,17]. Therefore, identifying suitable reference genes for specific contexts is crucial for precise gene expression analysis. *T. gondii*, an intracellular parasitic eukaryote, shares common reference genes with other eukaryotes, such as *actin*, *GAPDH*, and *tubulin*. Nevertheless, there is a lack of studies confirming the widespread and stable expression of these genes in *T. gondii*.

Based on the RNA-seq data of *T. gondii* subjected to varying concentrations of BRO, we identified stably expressed genes across different conditions to screen for suitable reference genes. The stability of these genes was assessed using two parameters: the coefficient of variation (CV) and the maximum fold change (MFC). Specifically, the MFC value for stably expressed genes must be less than 2, while the CV must be below 4% [18]. These two parameters indicate very low standard deviations and are widely recognized as essential criteria for selecting candidate reference genes [19]. Following these criteria, we identified seven stably expressed genes. Additionally, we selected three of the most commonly used reference genes in *T. gondii*. These genes are all genes encoding proteins involved in maintaining cellular functions.

In this study, we employed ΔCt, BestKeeper, geNorm, and NormFinder to evaluate candidate reference genes of *T. gondii* in response to varying concentrations of BRO. The results showed notable differences in reference gene stability as assessed by different software tools, likely due to variations in the algorithms used [20]. To address this, we utilized the online tool RefFinder to integrate the four algorithms and produce a final stability ranking. The findings reveal that at a BRO concentration of zero (specifically 0.1% DMSO), the most stable internal reference gene in *T. gondii* is *TGME49_205470*. Conversely, *TGME49_209030* (*ACT1*), a commonly used reference gene, was identified as the most unstable (Figure 5). For BRO concentrations of 1, 2, and 4 μg/mL, the most stable reference genes were inconsistent, specifically *TGME49_247220*, *TGME49_226020*, and *TGME49_205470*, respectively. When combining all data from each group for stability ranking, the results indicate the following order of stability under varying BRO concentrations: *TGME49_247220* > *TGME49_235930* > *TGME49_209030* > *TGME49_212300* > *TGME49_220950* > *TGME49_205470* > *TGME49_226020* > *TGME49_289690* > *TGME49_249180* > *TGME49_316400*.

Studies have shown that using multiple reference genes improves the accuracy of gene expression level assessments compared to relying on a single reference gene [16]. The geNorm software helps analyze pairwise variation (V value) to determine the optimal number of reference genes needed. Specifically, when the ratio V_n_/V_n+1_ is less than 0.15, it indicates that the appropriate number of reference genes is n. In this study, the V_2/V3_ values across various BRO treatment concentrations consistently fell below 0.15, suggesting that two reference genes are the most suitable.

To validate the selected reference genes, we compared the relative expression levels of six genes with the fragments per kilobase million (FPKM) values obtained from transcriptome analysis and found the results to be consistent. Additionally, we selected genes associated with autophagy, mitochondria, and lipid metabolism in *T. gondii* to conduct RT-qPCR experiments based on our previous research. Our results indicated that among the autophagy-related genes, BRO significantly decreased the expression of *ATG3* and *ATG7* while markedly increasing the expression of *ATG8*. The *ATG8* protein is a central component in the autophagy process of *T. gondii* and serves as the most widely used marker for autophagosomes [21,22]. Among the mitochondria-related genes, BRO significantly reduced the expression of *CYP450mt*, *ATPB*, and *ICAP2*. The *CYP450mt* enzyme is a steroidogenic enzyme localized in the mitochondria, and the CYP450mt gene is essential for the survival of *T. gondii* [23]. *ATPB* and *ICAP2* are subunits of *T. gondii* ATP synthase, playing a crucial role in ATP synthesis within the parasite [24]. Among lipid-related genes, BRO significantly reduced the expression of *ASH4* and *ACS1* while markedly increasing the expression of *DGAT*. The *ASH4* protein is an enzyme involved in lipid metabolism, and its absence can lead to the accumulation of phospholipids and neutral lipids within the *T. gondii* [25]. *ACS1* is closely associated with the neutral lipid metabolism of *T. gondii*. Following the deletion of *ACS1*, the level of lipid droplets in *T. gondii* increases [26]. The *DGAT* enzyme is a crucial triacylglycerol synthase and plays a significant role in the synthesis of neutral lipids [27,28]. Our results, consistent with previous studies, indicate that BRO can influence autophagy, mitochondrial function, and lipid synthesis in *T. gondii*.

In summary, under the influence of BRO, *TGME49_247220* and *TGME49_235930* were identified as the most appropriate reference genes for the RT-qPCR analysis of *T. gondii*.

## 4. Materials and Methods

### 4.1. Culture of T. gondii and Treatment with BRO

African green monkey kidney (Vero) cells were obtained from the cell bank of the Chinese Academy of Sciences, while RH tachyzoites were generously provided by the Lanzhou Veterinary Research Institute of the Chinese Academy of Agricultural Sciences. The Vero cells were cultured in DMEM (Gibco, Beijing China) supplemented with 10% fetal bovine serum (FBS, Gibco, Beijing China). *T. gondii* was cultivated in a monolayer of Vero cells in DMEM containing 3% FBS.

RH tachyzoites were added to a monolayer of Vero cells at a multiplicity-of-infection (MOI) ratio of 2:1. After 8 h, BRO was administered at concentrations of 4, 2, and 1 μg/mL, along with 0.1% DMSO for the control group, and the cells were incubated for 24 h. Following treatment, a cell scraper was used to detach the cells, and a 27 G needle was utilized to disrupt them three times in order to release the *T. gondii* tachyzoites. The mixture was centrifuged at 200× *g* for 5 min, and then the supernatant was passed through a 3 μm filter to remove cell debris. Finally, the supernatant was centrifuged again at 1500× *g* for 10 min to obtain the *T. gondii* pellet. Experiments were performed using three independent replicates. For the RNA-Seq data used in this article, the sample preprocessing method aligns with the aforementioned description.

### 4.2. Total RNA Extraction and Reverse Transcription

After washing the *T. gondii* pellet twice with phosphate-buffered saline (PBS, Solarbio, Beijing China), total RNA was extracted using RNAiso Plus (Takara, Beijing China) according to the manufacturer’s instructions. The concentration of the extracted RNA was measured using NanoDrop™ One (Thermo Scientific, Waltham, MA, USA), and the quality was assessed through agarose gel electrophoresis. After confirming satisfactory RNA quality, we used the PrimeScript™ RT Reagent Kit with gDNA Eraser (Takara, Beijing, China) to perform a reverse transcription reaction for complementary DNA (cDNA) synthesis.

### 4.3. Selection of Reference Genes and Primer Design

We selected ten genes for screening. Among these, *TGME49_209030* (*ACT1*) [29], *TGME49_316400* (*TUBA1*) [30], and *TGME49_289690* (*GAPDH1*) [31] are the most commonly used reference genes for *T. gondii*. Additionally, the genes *TGME49_220950*, *TGME49_205470*, *TGME49_235930*, *TGME49_212300*, *TGME49_226020*, *TGME49_249180*, and *TGME49_247220* were chosen based on their expression values (FPKM) from the *T. gondii* transcriptome (Table 5). These genes exhibit a low coefficient of variation (CV < 4%) and a narrow maximum fold change (MFC < 2) [32]. The MFC is defined as the ratio of the standard deviation of the FPKM values to the mean, while the CV is calculated as the ratio of the highest to the lowest FPKM values [18]. Primers were designed, and their specificity was assessed using NCBI’s Primer-BLAST tool. The primers were synthesized by Wuhan Qingke Biotechnology Company, and the primer sequences are listed in Table 6.

### 4.4. RT-qPCR Assay

The RT-qPCR reaction was performed using the TB Green^®^ Premix Ex Taq™ II kit (Takara, Beijing China), with cDNA from *T. gondii* treated with BRO and 0.1% DMSO at varying concentrations (4, 2, and 1 μg/mL) as the template. The cDNA concentration was measured and diluted to a starting concentration of 50 ng/μL. This cDNA was then further diluted in a 1:5 ratio to create six dilutions (50, 10, 2, 0.4, 0.08, and 0.016 ng/μL). The reaction mixture consisted of 10 μL of TB Green Premix Ex Taq II (Tli RNaseH Plus) (2×), 0.8 μL each of upstream and downstream primers (10 μM), 0.4 μL of ROX Reference Dye II (50×), 2 μL of the cDNA template, and 6 μL of ddH_2_O. A two-step PCR reaction was carried out using the QuantStudio™ 6 Flex Real-Time PCR system (Thermo Scientific, Waltham, MA, USA). The amplification program included an initial denaturation step at 95 °C for 30 s, followed by 40 cycles of 95 °C for 5 s and 60 °C for 34 s. Finally, a melting curve analysis was performed.

### 4.5. Stability Analysis of Candidate Reference Genes

The stability of the ten candidate reference genes was evaluated using the ΔCt method [13], BestKeeper [14], NormFinder [33], and GeNorm [15]. GeNorm was used to aggregate variation values and determine the optimal number of reference genes for normalization. To ensure the proper functioning of the four algorithms and to obtain a comprehensive stability ranking based on the experimental data, we employed the online tool RefFinder (https://blooge.cn/RefFinder/, accessed on 11 May 2024).

### 4.6. Verification of Reference Genes

Based on the transcriptome data, three upregulated and three downregulated genes were selected to validate the two most stable reference genes identified: *TGME49_247220* and *TGME49_235930*. Our previous studies have demonstrated that BRO can induce autophagy, mitochondrial dysfunction, and neutral lipid accumulation in *T. gondii* [11]. To further validate the stability of *TGME49_247220* and *TGME49_235930* as reference genes, we selected nine genes associated with autophagy, mitochondria, and lipid metabolism in *T. gondii* for RT-qPCR analysis. The primer sequences for these fifteen genes are provided in Appendix A. The RT-qPCR data were analyzed using the 2^−ΔΔCt^ method [34], with each experiment performed in triplicate.

## Figures and Tables

**Figure 1 ijms-25-11403-f001:**
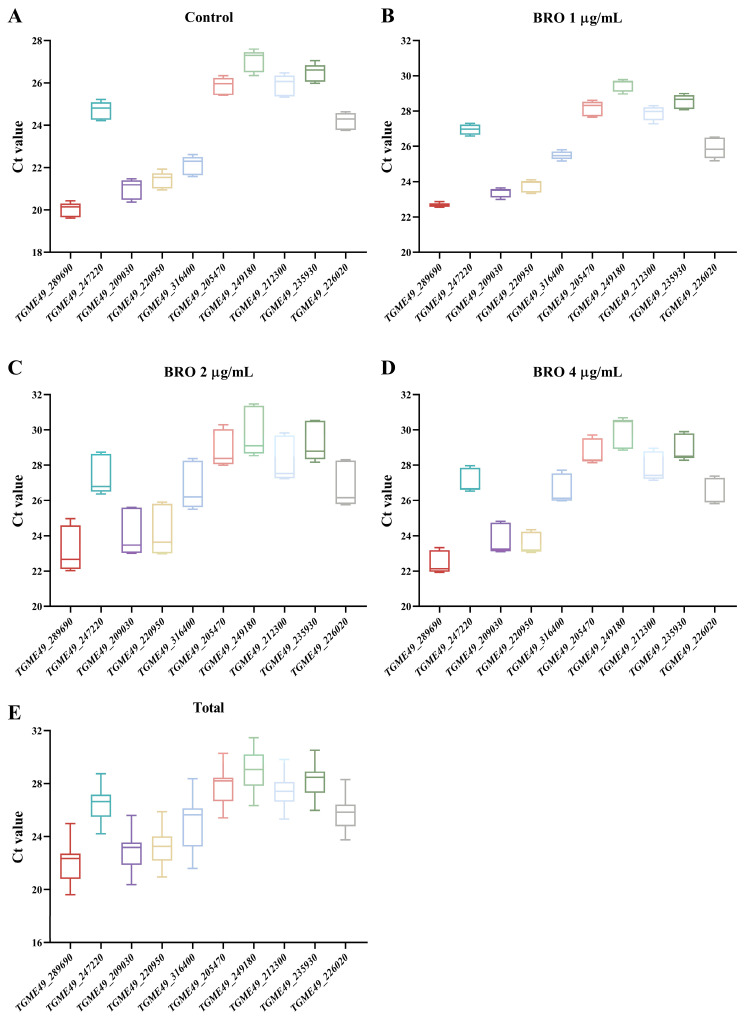
The expression levels of ten candidate reference genes were evaluated at different concentrations of broxaldine (BRO). (**A**) Control group with 0.1% DMSO; (**B**) BRO at 1 μg/mL; (**C**) BRO at 2 μg/mL; (**D**) BRO at 4 μg/mL; (**E**) all samples.

**Figure 2 ijms-25-11403-f002:**
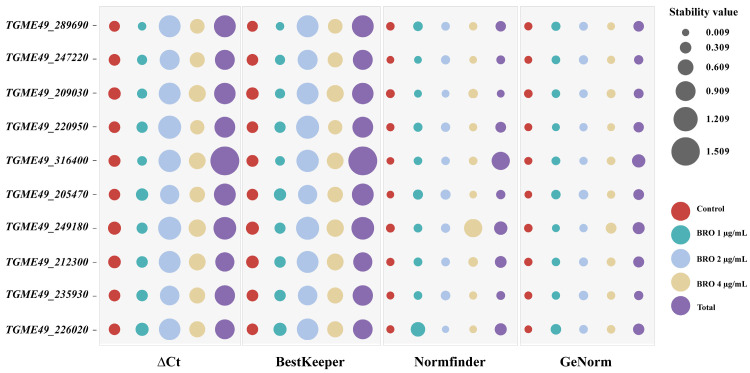
Assessment of the stability of ten candidate reference genes using four different methods—the ΔCt method, BestKeeper, NormFinder, and GeNorm—was performed. The size of each bubble represents the stability value, with larger bubbles indicating lower stability.

**Figure 3 ijms-25-11403-f003:**
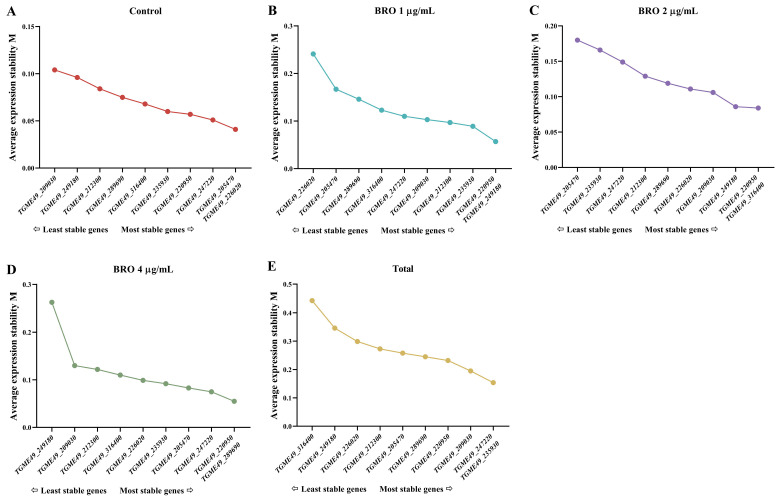
Average expression stability M of ten candidate reference genes under different concentrations of BRO. (**A**) Control group of 0.1% DMSO; (**B**) BRO 1 μg/mL; (**C**) BRO 2 μg/mL; (**D**) BRO 4 μg/mL; (**E**) all samples.

**Figure 4 ijms-25-11403-f004:**
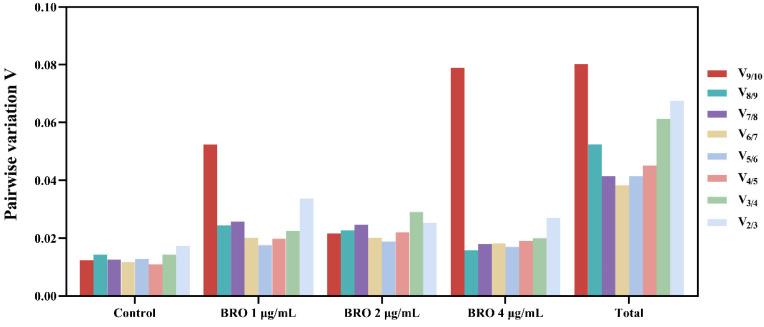
Pairwise variation values for all samples and groups.

**Figure 5 ijms-25-11403-f005:**
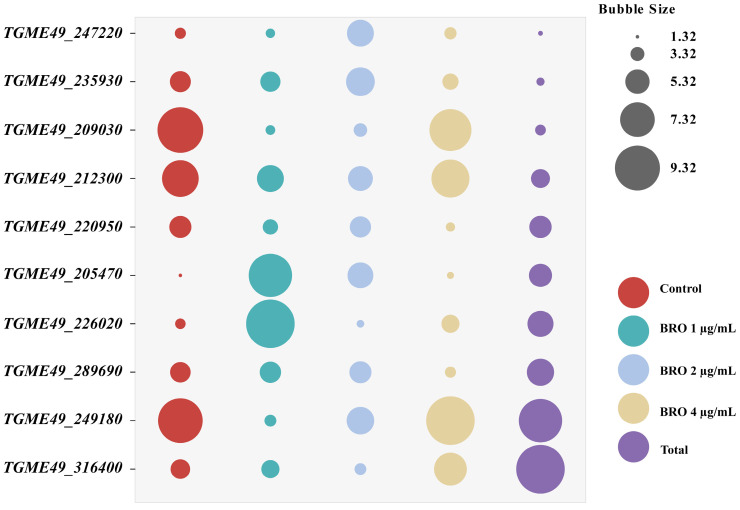
Comprehensive analysis of the stability of ten candidate reference genes using RefFinder. The size of the bubbles represents the geometric mean, with smaller bubbles indicating lower values and greater stability.

**Figure 6 ijms-25-11403-f006:**
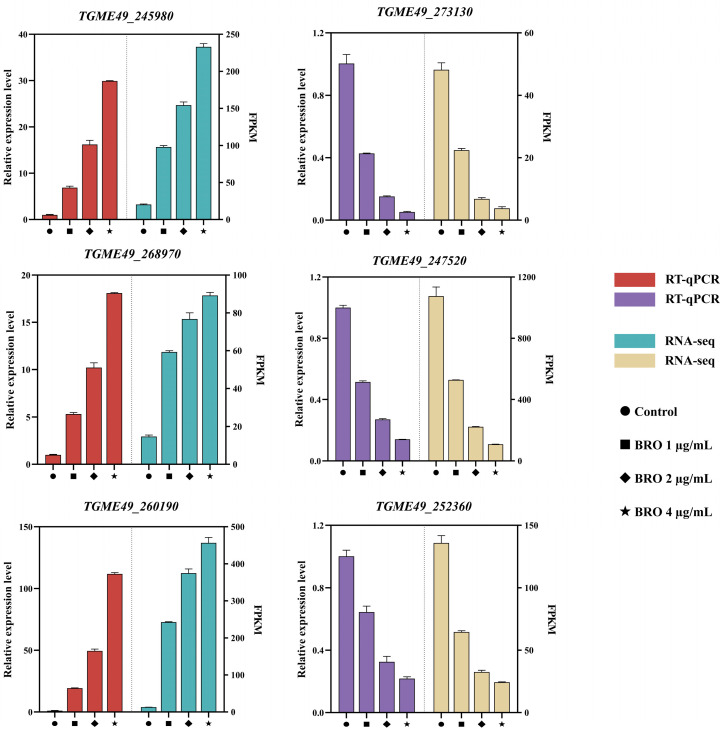
The expression levels of genes in RT-qPCR and RNA-seq experiments were assessed at varying concentrations of BRO. Data were analyzed using GraphPad Prism 9 software (San Diego, CA, USA). The results are presented as the mean ± SEM.

**Figure 7 ijms-25-11403-f007:**
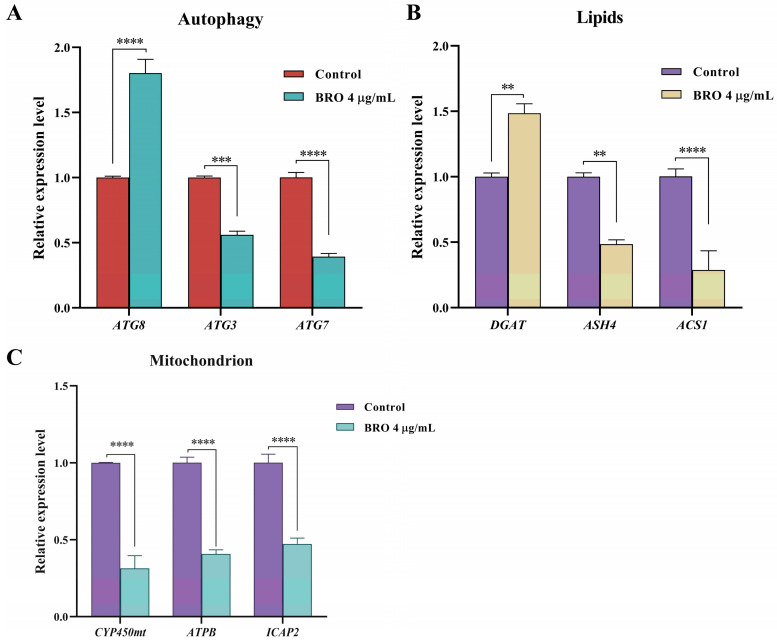
The expression levels of *T gondii* genes following treatment with 4 μg/mL BRO were assessed. Specifically, the expression levels of genes associated with autophagy (**A**), lipids (**B**), and mitochondria (**C**) in *T. gondii* were analyzed. Data were collected using GraphPad Prism 9 software, and the results are presented as the mean ± SEM. Statistical analysis was performed using Sidak’s two-way ANOVA with multiple comparisons. ****, *p* < 0.0001; ***, *p* < 0.001; **, *p* < 0.01.

**Table 1 ijms-25-11403-t001:** Amplification efficiency parameters of candidate reference genes.

Gene	Standard Curve	Efficiency	Correlation Coefficient (R^2^)
*TGME49_220950*	y = −3.553x + 26.75	91.18%	0.999
*TGME49_205470*	y = −3.319x + 30.79	100.12%	0.995
*TGME49_235930*	y = −3.462x + 31.07	94.47%	0.997
*TGME49_212300*	y = −3.445x + 32.96	95.11%	0.998
*TGME49_226020*	y = −3.364x + 31.19	98.27%	0.998
*TGME49_289690*	y = −3.243x + 31.46	103.40%	0.992
*TGME49_249180*	y = −3.507x + 33.73	92.82%	0.994
*TGME49_316400*	y = −3.428x + 30.92	95.76%	0.995
*TGME49_209030*	y = −3.276x + 28.69	101.95%	0.999
*TGME49_247220*	y = −3.217 + 32.36	104.57%	0.996

**Table 2 ijms-25-11403-t002:** Analysis of the mSD values of candidate reference genes using ΔCt method.

Gene	Control	BRO 1 μg/mL	BRO 2 μg/mL	BRO 4 μg/mL	Total
*TGME49_289690*	0.12	0.26	0.17	0.20	0.40
*TGME49_247220*	0.07	0.18	0.19	0.19	0.33
*TGME49_209030*	0.14	0.18	0.14	0.24	0.34
*TGME49_220950*	0.10	0.20	0.19	0.22	0.39
*TGME49_316400*	0.09	0.20	0.16	0.21	0.83
*TGME49_205470*	0.06	0.28	0.24	0.18	0.37
*TGME49_249180*	0.14	0.19	0.19	0.79	0.55
*TGME49_212300*	0.11	0.20	0.18	0.23	0.42
*TGME49_235930*	0.09	0.20	0.21	0.18	0.34
*TGME49_226020*	0.10	0.54	0.14	0.19	0.46

**Table 3 ijms-25-11403-t003:** BestKeeper was used to analyze the SD of candidate reference genes.

Gene	Control	BRO 1 μg/mL	BRO 2 μg/mL	BRO 4 μg/mL	Total
*TGME49_289690*	0.26	0.10	1.04	0.53	1.04
*TGME49_247220*	0.32	0.21	0.91	0.56	0.90
*TGME49_209030*	0.38	0.22	1.05	0.71	1.02
*TGME49_220950*	0.31	0.29	1.13	0.51	0.97
*TGME49_316400*	0.35	0.19	1.08	0.68	1.54
*TGME49_205470*	0.31	0.37	0.86	0.60	1.04
*TGME49_249180*	0.42	0.28	1.13	0.72	1.11
*TGME49_212300*	0.39	0.31	1.04	0.68	0.85
*TGME49_235930*	0.32	0.31	0.88	0.62	0.94
*TGME49_226020*	0.31	0.42	1.02	0.63	0.89

**Table 4 ijms-25-11403-t004:** The stability value of candidate reference genes was analyzed using NormFinder.

Gene	Control	BRO 1 μg/mL	BRO 2 μg/mL	BRO 4 μg/mL	Total
*TGME49_289690*	0.10	0.18	0.11	0.09	0.24
*TGME49_247220*	0.04	0.04	0.15	0.04	0.12
*TGME49_209030*	0.12	0.05	0.05	0.18	0.05
*TGME49_220950*	0.08	0.14	0.16	0.08	0.26
*TGME49_316400*	0.04	0.11	0.11	0.11	0.79
*TGME49_205470*	0.03	0.21	0.21	0.01	0.15
*TGME49_249180*	0.12	0.11	0.15	0.79	0.44
*TGME49_212300*	0.08	0.12	0.12	0.18	0.33
*TGME49_235930*	0.05	0.11	0.18	0.07	0.12
*TGME49_226020*	0.07	0.52	0.01	0.04	0.35

**Table 5 ijms-25-11403-t005:** Transcriptome FPKM, CV, and MFC of candidate reference genes for *T. gondii*.

Gene	FPKM				CV (%)	MFC
	Control	BRO 1 μg/mL	BRO 2 μg/mL	BRO 4 μg/mL		
*TGME49_220950*	795.5	809.98	810.17	828.2	1.65	1.04
*TGME49_205470*	255.29	263.88	269.57	277.13	3.46	1.09
*TGME49_235930*	170.62	177.18	182.85	182.87	3.26	1.07
*TGME49_212300*	247.93	257.77	260.49	268.05	3.22	1.08
*TGME49_226020*	365.64	372.79	381.07	391.27	2.92	1.07
*TGME49_247220*	545.4	569.4	583.65	594.77	3.72	1.09
*TGME49_249180*	58.97	60.57	63.02	63.5	3.46	1.08
*TGME49_316400* (*TUBA1*)	1180.95	594.29	318.97	264	71.22	4.47
*TGME49_209030* (*ACT1*)	513.44	402.78	304.58	294.56	26.97	1.74
*TGME49_289690* (*GAPDH1*)	390.98	311.95	289.81	384.62	14.83	1.35

**Table 6 ijms-25-11403-t006:** Information on candidate reference genes and primer sequences for *T. gondii*.

Gene	Gene Description	Primer Sequences (5′→3′)	Product Size (bp)
*TGME49_220950*	mitochondrial association factor 1 (*MAF1*)	F: CGGCAACCTGAACAACAACG R: CCTTGCACTGGGTACTGCTG	162
*TGME49_205470*	translation elongation factor 2 family protein, putative	F: ATCATGGACCCGATCTGCAC R: TCCCTGTCGTCACCCTTGA	100
*TGME49_235930*	domain K- type RNA binding proteins family protein	F: TATCCTTGGCTCTGGCGGT R: GCTGCATGACGAAACCGATG	158
*TGME49_212300*	dense granule protein *GRA32*	F: GGAATCGGAAGGGGCGTATT R: GCAGGGCTTGGAACTTGTTG	72
*TGME49_226020*	transporter, major facilitator family protein	F: TGCTTGCGGGATATTGGCT R: TGCGAAGTAGCCTCCCATTG	125
*TGME49_289690*	glyceraldehyde-3-phosphate dehydrogenase *GAPDH1*	F: ATTTTGCTTGGGATTCGAGGA R: TGCAGGGTAACGATCAAAAAATG	93
*TGME49_249180*	bifunctional dihydrofolate reductase-thymidylate synthase	F: CAGACTACACAGGTCAGGGC R: CACAACAAGTGACAAGGCGG	145
*TGME49_316400*	alpha tubulin *TUBA1*	F: GCCAAGTGTGATCCTCGTCA R: GGCTGGTAGTTGATACCGCA	170
*TGME49_209030*	actin *ACT1*	F: TCGGAATGGAGGAGAAGGACTGC R: AGTTCGTTGTAGAAGGTGTGATGCC	148
*TGME49_247220*	udix-type motif 9 isoform a family protein	F: AATGGGAGACTTCAGGTGGC R: GCGTAACTATGAGCGGTCCA	106

## Data Availability

The NCBI database provides access to high-throughput data. The accession number associated with the RNA-seq data presented in this study is PRJNA1153788.

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
