# Peer review of "Reference Gene Selection for RT-qPCR Normalization in Toxoplasma gondii Exposed to Broxaldine"

_ijms, 2024, doi:10.3390/ijms252111403_

Round 1
Reviewer 1 Report
Comments and Suggestions for Authors
Review: Yanhua Qiu, Yubin Bai, Weiwei Wang, Qing Wang, Shulin Chen, and Jiyu Zhang, “Reference gene selection for RT-qPCR normalization in Toxoplasma gondii exposed to broxaldine ”
This group previously profiled the activity of Broxaldine, a compound that inhibits host cell invasion, intracellular replication and differentiation of Toxoplasma in vitro. Treatment induces mitochondrial swelling and enhances cytoplasmic vacuolization, with parasite fragmentation. Both tachyzoite and bradyzoite forms had TEM evidence for autophagic lysosomes. Therefore, this group hypothesizes that Broxaldine induces autophagy, perhaps as a consequence of mitochondrial damage (indicated by the disappearance of mitochondrial cristae after treatment). In human cells, Broxaldine induces expression of heme-oxygenase-1, catalyzing the rate-limiting step of heme degradation.
The current manuscript concerns developing methods for RT-qPCR target gene expression measurements relative to appropriate reference gene expression.
1. There is no discussion of how the candidate reference genes were selected nor a consistent definition of what they encode. This justification will be important to readers.
2. While the paper develops evidence for the best reference genes to use, there is no further information to illustrate their use in an experiment. One possible solution would be to use the best reference genes to evaluate expression of genes for mitochondrial enzymes (such as those for heme biosynthesis), lipid synthesis and autophagy. I know that this is the likely next step, but I think completion of control experiments alone does not merit publication as a paper in this form.
3. I would also recommend that the authors look at expression changes in their host cells with Broxaldine treatment as Heme oxygenase-1 activity is involved in the control of Toxoplasma gondii infection in the lung of BALB/c and C57BL/6 and in the small intestine of C57BL/6 mice (see Ester CB Araujo paper).
Comments on the Quality of English Language
I did not detect significant problems with writing.
Author Response
Thank you very much for taking the time to review this manuscript. Please find the detailed responses below and the corresponding revisions/corrections highlighted/in track changes in the re-submitted files
Comment 1: There is no discussion of how the candidate reference genes were selected nor a consistent definition of what they encode. This justification will be important to readers.
Response 1: Thank you for pointing this out. The criteria for selecting reference genes are detailed in Section 4.3 of lin260. We selected seven genes for the screening of reference genes in accordance with the standards established by Carvalho et al. Additionally, we included the three most commonly used internal reference genes in T. gondii for this screening. Detailed coding information for these genes is provided in Table 6 (line 274). The descriptions of the two genes, TGME49_220950 and TGME49_212300, in the table are based on data from the T. gondii website (https://toxodb.org/toxo/app/), which has been updated with additional details. In our discussion, we have incorporated the following description in line 178.
“Based on the RNA-seq data of T. gondii subjected to varying concentrations of BRO, we identified stably expressed genes across different conditions to screen for suitable reference genes. The stability of these genes was assessed using two parameters: the coefficient of variation (CV) and the maximum fold change (MFC). Specifically, the MFC value for stably expressed genes must be less than 2, while the CV must be below 4%. These two parameters indicate very low standard deviations and are widely recognized as essential criteria for selecting candidate reference genes. Following these criteria, we identified seven stably expressed genes. Additionally, we selected three of the most commonly used reference genes in T. gondii. These genes are all genes encoding proteins involved in maintaining cellular functions.”
Comment 2: While the paper develops evidence for the best reference genes to use, there is no further information to illustrate their use in an experiment. One possible solution would be to use the best reference genes to evaluate expression of genes for mitochondrial enzymes (such as those for heme biosynthesis), lipid synthesis and autophagy. I know that this is the likely next step, but I think completion of control experiments alone does not merit publication as a paper in this form.
Response 2: We have incorporated relevant experiments, and the associated changes are detailed in the methods (line 297), results (line 156), discussion (line 210), and supplementary materials.
Comment 3: I would also recommend that the authors look at expression changes in their host cells with Broxaldine treatment as Heme oxygenase-1 activity is involved in the control of Toxoplasma gondii infection in the lung of BALB/c and C57BL/6 and in the small intestine of C57BL/6 mice (see Ester CB Araujo paper).
Response 3: Your opinion is very important, thank you very much. We agree with your comment. Your suggestion provides a direction for our next research. Unfortunately, due to the limited time, we did not supplement experimental. In this study, we aimed to explore the stable reference genes of T. gondii under the action of BRO. Therefore, we mainly focus on the results of reference gene screening of T. gondii.
Reviewer 2 Report
Comments and Suggestions for Authors
Authors have demonstrated in their previous study that broxaldine (BRO) has antiparasitic activity on both tachyzoites and bradyzoites of Toxoplasma gondii, potentially affecting the parasite’s autophagy, mitochondrial function, and neutral lipid synthesis. In this paper, the authors optimized RT-qPCR assays to assess the expression of target genes in T. gondii treated with various concentrations of BRO. A key challenge when analyzing target gene expression relative to reference or housekeeping genes is ensuring that the expression of these reference genes remains stable under treatment conditions. To address this, the authors used the ΔCt method, BestKeeper, geNorm, and NormFinder to evaluate potential reference genes in response to different concentrations of BRO. RefFinder was employed to integrate the results of these algorithms, generating a final stability ranking. TGME49_247220 and TGME49_235930 were identified as the most stable reference genes for RT-qPCR analysis of T. gondii under BRO treatment.
Comments: The findings from these studies are crucial for evaluating the relative expression of target genes in comparison to reference control genes. However, gene expression can also be measured absolutely (by using gene-specific DNA products and generating a standard curve against copy number) under specific treatment conditions, without the need for reference genes. This limits the scope of the results to some extent. Nevertheless, the study clearly demonstrates that the reference genes identified by the authors will be valuable for relative gene expression studies. The authors could enhance the introduction by citing examples where relative gene expression studies in Toxoplasma have been hindered by the criteria used for selecting reference genes.
Line 44: Transcriptome data suggest that BRO may alter the expression of commonly used reference genes in T. gondii: Are the authors referring to a whole-genome transcriptome analysis of the parasites treated with Broxaldine? If so, could you please provide the reference for that?
The authors mentioned using RNA-Seq data to assess gene copy numbers, but it's unclear whether these datasets were generated from parasites treated with Broxaldine. Could you clarify if that's the case?
Author Response
Thank you very much for taking the time to review this manuscript. Please find the detailed responses below and the corresponding revisions/corrections highlighted/in track changes in the re-submitted files
Comment 1: The findings from these studies are crucial for evaluating the relative expression of target genes in comparison to reference control genes. However, gene expression can also be measured absolutely (by using gene-specific DNA products and generating a standard curve against copy number) under specific treatment conditions, without the need for reference genes. This limits the scope of the results to some extent. Nevertheless, the study clearly demonstrates that the reference genes identified by the authors will be valuable for relative gene expression studies. The authors could enhance the introduction by citing examples where relative gene expression studies in Toxoplasma have been hindered by the criteria used for selecting reference genes.
Response 1: Thank you for pointing this out. In fact, after discovering during the experiment that BRO influences the expression of commonly used reference genes of T. gondii, we conducted an extensive literature review to identify relevant studies on these internal reference genes. Unfortunately, we found almost no pertinent information. We have included the following description in the preface (line 37).
“In RT-qPCR experiments, researchers typically select widely used reference genes for normalizing gene expression. However, numerous studies have shown that these commonly used reference genes, such as GAPDH and β-Actin, display variable expression levels under different conditions.”
Comment 2: Line 44: Transcriptome data suggest that BRO may alter the expression of commonly used reference genes in T. gondii: Are the authors referring to a whole-genome transcriptome analysis of the parasites treated with Broxaldine? If so, could you please provide the reference for that?
Response 2: Broxaldine (BRO) may influence the expression of commonly used reference genes in T. gondii. This conclusion is based on RNA-seq data obtained after treating T. gondii with various concentrations of BRO. The dataset has been uploaded to the NCBI database. The gene expression levels (FPKM values) of the commonly used reference genes TGME49_316400 (TUBA1), TGME49_209030 (ACT1), and TGME49_289690 (GAPDH1) under the influence of different concentrations of BRO are presented in Table 5 (Line 273) of the article. We apologize for not clearly distinguishing the names of the commonly used internal reference genes in the table, which was an oversight on our part. We have since corrected this in the text.
Comment 3: The authors mentioned using RNA-Seq data to assess gene copy numbers, but it's unclear whether these datasets were generated from parasites treated with Broxaldine. Could you clarify if that's the case?
Response 3: This oversight was an error on our part, and we sincerely appreciate the review experts for bringing this issue to our attention. The RNA-seq data used in this article were all obtained from T. gondii following BRO treatment. The treatment of T. gondii is identical to the processing method described in this article. We have added the following clarification to the article (line 250).
“For the RNA-Seq data used in this article, the sample preprocessing method aligns with the aforementioned description.”
Round 2
Reviewer 1 Report
Comments and Suggestions for Authors
I am happy with the changes that the authors made to the revised manuscript.
Comments on the Quality of English LanguageI am happy with the changes that the authors made to the revised manuscript.
Author Response
Comment 1: I am happy with the changes that the authors made to the revised manuscript.
Response 1: We are very flattered by your confirmation.